# Sarcosine May Induce EGF Production or Inhibit the Decline in EGF Concentrations in Patients with Chronic Schizophrenia (Results of the PULSAR Study)

**DOI:** 10.3390/ph16111557

**Published:** 2023-11-03

**Authors:** Agnieszka Pawlak, Bartosz Kaczmarek, Adam Wysokiński, Dominik Strzelecki

**Affiliations:** 1Department of Affective and Psychotic Disorders, Medical University of Łódź, ul. Czechosłowacka 8/10, 92-216 Łódź, Poland; bartosz.kaczmarek@mp.pl; 2Department of Old Age Psychiatry and Psychotic Disorders, Medical University of Łódź, ul. Czechosłowacka 8/10, 92-216 Łódź, Poland; adam.wysokinski@umed.lodz.pl

**Keywords:** EGF, sarcosine, schizophrenia, negative symptoms

## Abstract

Sarcosine (N-methylglycine), a glutamatergic modulator, reduces the primary negative symptoms of schizophrenia. These beneficial changes might be mediated by trophic factors such as epidermal growth factor (EGF). We assessed associations between initial serum EGF levels or changes in serum EGF levels and symptom severity during the addition of sarcosine to stable antipsychotic treatment and thereby evaluated the associations between glutamatergic modulation, clinical changes and peripheral EGF concentrations. Fifty-eight subjects with a diagnosis of chronic schizophrenia with dominant negative symptoms, stably treated with antipsychotics, completed a prospective 6-month, randomized, double-blind, placebo-controlled study. Subjects received orally 2 g of sarcosine (*n* = 28) or placebo (*n* = 30) daily. Serum EGF levels and symptom severity (using the Positive and Negative Syndrome Scale (PANSS) and the Calgary Depression Scale for Schizophrenia (CDSS)) were assessed at baseline, 6-week and 6-month follow-up. Augmentation antipsychotic treatment with sarcosine had no effect on EGF serum levels at any time points. Only the sarcosine group showed a significant improvement in negative symptoms, general psychopathology subscales and the overall PANSS score. We found a reduction in serum EGF levels in the placebo group, but levels in the sarcosine remained stable during the study. Our data indicate that improvement in negative symptoms due to sarcosine augmentation is not directly mediated by EGF, but effective treatment may induce the production or block the decrease in EGF concentrations, which indicates the neuroprotective effect of treatment and confirms the relationship between neuroprotection and EGF levels.

## 1. Introduction

Focus on the importance and role of growth factors in schizophrenia has been steadily increasing. Their properties—influence on neuronal and glial growth, differentiation, migration, functioning and survival—seem to have scientific and clinical potential. It is known that changes in the cytoarchitecture of the nervous tissue, as well as cellular dysfunctions, e.g., changes in metabolism, morphology, interconnections and receptors, are of great importance for the onset and progression of schizophrenia and negative symptoms in particular [1,2].

The epidermal growth factor (EGF) is extensively distributed in the brain tissue (e.g., in the prefrontal cortex, hippocampus, putamen and amygdala) and can be found in neurons and astrocytic glia. Binding to its receptor (EGRF) on the cell surface, EGF activates the intrinsic EGRF protein-tyrosine kinase activity, which begins the cascade of signal transduction (an increase in cellular calcium levels, synthesis of protein, glycolysis and gene expression), leading to cell proliferation [3]. The literature on the relationship between EGF and schizophrenia is sparse, but researchers indicate that EGF is associated with deficits in sensorimotor gating, latent inhibition, working memory, impaired social interaction and overall schizophrenia severity [4,5,6]. Perinatal exposure to EGF in rodents results in an irreversible hyperdopaminergic state [7], which is particularly interesting. For this to be the case, we should have more data on the involvement of these factors in the pathophysiological processes relevant to the various symptom dimensions of schizophrenia and other psychiatric disorders; meanwhile, as we noted above, current data on this subject are scarce. The results of several published studies do not provide clear conclusions. In older studies, serum EGF levels are elevated in schizophrenia [5,8,9], whereas more recent studies indicate a reduction in EGF levels in the first episode [10] or in both the first episode and chronic schizophrenia [11]. We are particularly interested in research on chronic schizophrenia, which we define here as a form with marked negative symptoms and impaired functioning, e.g., cognitive and social, with a relatively low intensity of positive symptomatology. It is worth noting, that EGF is utilized in clinical practice, e.g., in Parkinson’s disease or diabetic foot ulcers; it then might be a very attractive therapeutic option in various neuropsychiatric conditions [12,13,14,15]. 

Glutamatergic and GABAergic (excitatory and inhibitory, respectively) systems are involved in the pathogenesis of negative, positive, cognitive and affective symptoms in schizophrenia. In the study, we assess the effects of sarcosine (N-methylglycine), a glycine transporter type I (GlyT1) inhibitor, located on astroglia [16,17]. Sarcosine causes an increase in levels of NMDA (N-methyl-D-aspartate) co-agonist glycine around the receptor and thus enhances the activity of this receptor, particularly on GABAergic inhibitory interneurons whose function in schizophrenia is disturbed [18,19]. Previous sarcosine trials in schizophrenia showed its beneficial impact on negative symptoms’ severity and showed better results in this area than conventional therapy [20,21]. While negative and cognitive symptoms are closely linked with patients’ functioning and quality of life level, we need options to help patients recover as much as possible. Data analysis in our population (the PULSAR study—Polish Sarcosine Study in Schizophrenia) indicates significant sarcosine-induced changes in different aspects of symptomatology: scores in negative symptoms, general psychopathology subscales and total Positive and Negative Syndrome Scale score, with simultaneous positive changes in neuronal or glial parameters in left dorsolateral prefrontal cortex, left hippocampus and left frontal white matter in magnetic resonance spectroscopy when compared to placebo [22,23,24,25]. 

Given our special interest in negative symptoms during the design of the study, we placed particular emphasis on examining the possible relationship between the severity of these symptoms (avolition, anhedonia, antisociality, flat affect, alogia) and the blood concentrations of EGF which might be related to negative symptoms and cognitive impairment in this psychosis, which may be indicated by the results of research in Parkinson’s and Alzheimer’s diseases [4,5,6,26,27]. The connection of the EGF family with the pathogenesis of schizophrenia is particularly well documented in the case of neuregulin-1, structurally very similar to EGF. It is known that neuregulin-1 is both associated with schizophrenia and glutamatergic transmission [28,29,30]. If EGF is involved in the mechanism of action of glutamatergic modulators (directly or indirectly), this could amend the picture of the pathomechanisms underlying schizophrenia and open new therapeutic possibilities. 

Both first- and second-generation antipsychotics are the recognized first-line treatment for patients with schizophrenia. Their efficacy is mainly limited to positive symptoms (delusions, hallucinations and thought disorders), while negative symptoms do not respond well to currently used drugs or forms of therapy. Stopping psychotic symptoms was previously considered a success, but there is a great need to improve other dimensions of psychosis such as primary negative, cognitive and affective symptoms, and thus overall quality of life and functioning.

Previous studies have considered the severity of all schizophrenia symptoms and have not elucidated whether EGF levels are related to the severity of positive, negative and affective dimensions. To provide more accurate results, biochemical measurements and detailed anthropometric measurements were combined with body composition analysis, determined via bioelectrical impedance analysis (BIA), which provides accurate measurements of body fat and lean body mass. 

Our aim is to determine whether EGF levels in peripheral blood are associated with the severity of schizophrenia symptoms and whether modulation of glutamatergic transmission has an impact on both symptomatology and EGF concentrations. 

Based on our results and the available literature, sarcosine reduces the severity of negative symptoms. Therefore, as primary outcomes, we want to observe whether EGF concentrations change during the study and whether they are correlated with PANSS negative subscale scores during the 6-month administration of the amino acid. We preliminarily assume that they should increase during sarcosine administration. 

As secondary outcomes, we also want to evaluate possible correlations of EGF concentrations with the severity of positive dimensions, overall psychopathology, total PANSS score and affective symptoms assessed with the Calgary Depression Scale for Schizophrenia (CDSS). 

To the best of our knowledge, this is the first study to investigate a similar combination of parameters in subjects with schizophrenia.

## 2. Results 

Sixty patients were randomly assigned to receive sarcosine (*n* = 30) or placebo (*n* = 30) and completed the 6-month, double-blind, placebo-controlled study. Two patients in the sarcosine group did not complete blood tests for EGF. Therefore, 58 patients were analyzed, including 28 patients taking sarcosine (Figure 1). 

Both groups were very comparable in terms of demographic, clinical, therapeutic, anthropometric, and metabolic parameters (Table 1).

All participants remained on stable antipsychotic treatment throughout the study. Clinically, there were no differences between baseline PANSS scores in our two subgroups. However, patients in the sarcosine group achieved a significant improvement in PANSS scores at the end of the study compared to the placebo group. We noted a reduction in negative subscale score: −6.7 ± 3.5 vs. −0.8 ± 1.7, *p* < 0.001; a reduction in general subscore: −5.7 ± 5.7 vs. −1.4 ± 5.6, *p* = 0.006; a reduction in total score: −13.2 ± 8.3 vs. −2.3 ± 8.7, *p* < 0.001. There was no significant change in PANSS positive subscale score (−0.6 ± 2.7 vs. −0.1 ± 3.4, *p* = 0.4) or CDSS score between the two groups (sarcosine: −0.2 ± 1.8, placebo: 0.9 ± 3.3, *p* = 0.2). At the end of the study, there were no significant changes in any of the analyzed cardiometabolic and body composition parameters in both groups.

Figure 2 shows EGF levels in the study groups at three time points—initially, after six weeks and after six months. At the beginning of the study, EGF levels were comparable in both study groups (sarcosine: 235.1 ± 165.0 pg/mL, placebo: 315.5 ± 226.4 pg/mL, *p* = 0.2). There was a significant difference in EGF levels (sarcosine: 200.7 ± 117.9 pg/mL, placebo: 116.1 ± 72.6 pg/mL, *p* = 0.03) after six weeks, but not after six months (sarcosine: 288.1 ± 231.3 pg/mL, placebo: 182.1 ± 135.8 pg/mL, *p* = 0.1). 

Next, we analyzed changes in EGF levels between the study groups at each of the three time points; see Figure 3.

These changes were found to be non-significant for initial, 6-week and 6-month time points. There was no significant change in EGF after six weeks or six months in the sarcosine group. Interestingly, there was a significant and progressing decrease between baseline EGF levels after six weeks (*p* = 0.02) and after six months (*p* = 0.01) in the placebo group. Initial EGF levels were correlated with levels at six weeks (rho = 0.38, *p* = 0.04), change after six weeks (rho = −0.56, *p* = 0.001) and after six months (rho = −0.59, *p* < 0.001). There were no correlations between initial EGF level and PANSS or CDSS scores/subscores including negative symptoms PANSS subscale. Also, there were no correlations between changes (initial vs. after six months) in EGF level and changes in PANSS or CDSS scores/subscores. Initial EGF levels were not correlated with the previous clinical course of schizophrenia and its treatment (treatment duration, number of hospitalizations, antipsychotics dose), nor with cardiometabolic parameters. Also, there was no difference in initial or change (initial vs. six months) levels of EGF between patients whose scores in PANSS total, P/N/G subscores and CDSS score improved or did not.

## 3. Discussion

This is the first randomized placebo-controlled study examining the impact of sarcosine on serum EGF concentration in schizophrenia with predominant negative symptoms.

The first primary study objective was to assess changes in EGF serum levels during antipsychotic treatment augmentation with sarcosine. We found that EGF levels were stable throughout the project in the sarcosine group, while there was a significant decrease in EGF levels in the control group despite no differences between the two groups at baseline. 

The second primary objective of the study was to test correlations of EGF concentrations with PANSS negative subscale scores in the study group. There were no correlations between initial EGF level and negative symptoms PANSS subscale scores and between changes (initial vs. after six months) in EGF level and changes in negative symptoms PANSS subscale scores.

The secondary study objective was to evaluate whether EGF levels correlate with the severity of positive symptoms, overall psychopathology, total PANSS score and affective symptoms of schizophrenia and its changes. 

We found no correlations between initial EGF level and PANSS total/positive/general psychopathology and CDSS scores. Also, there were no correlations between changes (initial vs. after six months) in EGF level and changes in PANSS total/positive/general psychopathology or CDSS scores.

Our study outcomes demonstrate that EGF does not directly relate to the symptoms’ severity and improvement in PANSS (total and subscale scores) and CDSS scales during sarcosine use. We conclude that the clinical effect achieved during sarcosine treatment is not mediated by EGF, at least not as indicated by changes in peripheral blood concentrations. The clinical difference is not due to mental deterioration in the control group, which could link negative symptomatology to EGF levels. We assume that sarcosine and potentiation of NMDA receptor function also have no direct effect on changes in peripheral EGF concentrations.

However, observations on EGF concentrations in the sarcosine and placebo groups may indicate that the neuroprotective effect of sarcosine (by normalizing glutamatergic transmission and reducing its excitotoxic effects) may induce EGF production or inhibit the decline in EGF concentrations that progressed in the placebo group. EGF is probably the only growth factor that also has neuroprotective effects—protecting dopaminergic neurons from the excitotoxic effects of elevated glutamate concentrations [31,32] and inducing reparative processes [27,33,34,35]. Morphological changes induced by susceptibility to schizophrenia, as well as toxic processes in the course of acute psychosis, have a particularly negative impact on the increasing level of negative symptoms and cognitive dysfunctions [36,37,38]. In schizophrenia, GlyT1 blockade on GABAergic neurons prevents glutamatergic excitotoxicity, i.e., sarcosine can block neuronal and glial damage/dysfunction, which can also prevent EGF decline [39]. The bioavailability of EGF in the brain is ensured by production in glial cells and neurons and by uptake from the peripheral circulation [3], indicating that peripheral concentrations may be somewhat related to levels within the brain, but this is not described in detail yet. It may, of course, affect the interpretation of our results, especially since EGF is not specific to neural tissue and is found, for example, in the skin, salivary glands and platelets [40,41]. 

Our study has limitations: relatively small study groups and heterogeneous antipsychotic treatment across the study groups. On the other hand, in both study groups, the statuses of psychosis, anthropometric and metabolic parameters were comparable, limiting the potential impact of these co-variables on EGF levels. The exception is tobacco smoking (the percentage of smokers was significantly higher in the placebo group), which has potential importance, as smoking is known to reduce EGF expression. However, the above findings relate to salivary glands, but there are no data on levels in the brain [42].

## 4. Materials and Methods

### 4.1. Participants and Study Design

All enrolled patients were European Caucasians aged 18–60 years with a diagnosis of paranoid schizophrenia (295.30, according to DSM-IV, F20.0 according to ICD-10). Recruitment was conducted in our psychiatric outpatient clinic. Patients in stable physical, neurological and endocrine condition with laboratory values in the normal range (blood and biochemical tests including TSH, hepatic and renal parameters) and normal electrocardiogram were eligible for enrollment. After precise information about the aims and methods of the study and prior to all subsequent procedures, all eligible patients signed the informed consent form. After consent, they underwent a structured interview in accordance with the criteria of schizophrenia by ICD-10 and DSM-IV. One of the main inclusion criteria was the stability of the mental state and predominance of negative symptoms, with at least a 3-point score in each item in the PANSS negative symptoms subscale. Patients with symptoms of psychotic exacerbation, i.e., scoring more than 3 on one of the items in the positive symptoms subscale, were excluded from participation. To evaluate the influence of sarcosine precisely, we required 3-month-long treatment dosage stability before enrollment. Patients declaring suicide risk or taking clozapine (combining glutamatergic modulators such as glycine or sarcosine with clozapine was not effective or led to worsening of mental state by increasing positive symptoms) also did not enter the trial [43,44]. 

Project PULSAR was designed as a 6-month parallel-group, randomized, double-blind, placebo-controlled study. Participants were randomly assigned to the sarcosine (*n* = 30) or placebo (*n* = 30) groups in a 1:1 ratio using a protocol obtained from a designated website (http://www.randomization.com), accessed 6 October 2012. The randomization process was conducted by a person not involved in the patient assessments. The randomization code was not broken until all patients had completed the study. We added both sarcosine and placebo to ongoing and stable antipsychotic treatment. All patients received plastic capsules containing 2 g of the amino acid or microcrystalline cellulose as a placebo. Participants were then trained to open the container, dissolve the contents in water and drink the solution once a day in the morning.

No financial industry was involved in the study. For more information on the Polish Sarcosine Study in Schizophrenia (PULSAR), visit ClinicalTrials.gov, study identifier: NCT01503359.

### 4.2. Measurements

All blood measurements were taken at least twice—at the visit before the start of the sarcosine or placebo, and after six months—following the last dose of study treatment. In 14 subjects in the placebo group and in 15 subjects in the sarcosine group, we arranged additional assessments of EGF levels after the first six weeks. 

#### 4.2.1. Clinical Evaluation 

Evaluators assessed schizophrenia symptoms using the PANSS scale (total score and subscales of positive, negative and general symptoms); to assess the severity of depressive symptoms, we used the Calgary Depression Scale for Schizophrenia (CDSS). For each patient, all assessments on all scales were conducted continuously by a single trained investigator. CDSS scores above 6 points classified patients as depressive.

#### 4.2.2. Blood

Blood was sampled and then immediately transferred to our laboratory between 7 a.m. and 8 a.m., after at least 8 h of fasting. The material was centrifuged using 3500 rpm at 22 °C for 10 min. Serum EGF levels were established following the kit instructions using commercially available high-sensitivity ELISA plates (Diaclone, Besancon Cedex, France), intra-test CV < 4.5%, inter-test CV < 9.2%. Before ELISA evaluation, serum samples were kept at −80 °C. The optical density of the wells was measured at the Central Scientific Laboratory of the Medical University of Lodz using an automated microplate reader (Emax; Molecular Devices, San Jose, CA, USA). Lipid serum and glucose levels were analyzed using the Dirui CS-400 device (Dirui, Changchun, China).

#### 4.2.3. Anthropometry

Patients’ height was measured to the nearest 0.5 cm using a wall-mounted height tape measure. Weight was measured and recorded to the nearest 0.5 kg in light clothing without shoes using a spring balance placed on a horizontal stable surface. Body mass index (BMI) was calculated as weight (kg)/height (m)^2^. We also measured waist, abdomen and hip circumference using a non-stretch measuring tape.

#### 4.2.4. Body Composition

We used the Maltron BF-906 body fat analyzer (Maltron, Rayleigh, UK), a single-frequency bioelectrical impedance analyzer (BIA), to measure total body fat and lean body mass and determine resistance and reactance at 50 Hz. BIA assessments were performed by trained personnel. In a short summary, BIA determines the electrical impedance or opposition to the flow of electrical current through body tissues, allowing estimation of total body water and then (using Maltron’s proprietary equations) lean body mass and body fat.

#### 4.2.5. Determination of Metabolic Syndrome and Other Measurements

We implemented the International Diabetes Foundation (IDF) criteria for metabolic syndrome and abdominal obesity. Impaired fasting glucose was defined as a fasting plasma glucose concentration ≥ 100 mg/dL. Dyslipidemia was diagnosed when triglycerides (TGA) were ≥150 mg/dL and/or total cholesterol (TC) ≥ 200 mg/dL and/or reduced HDL cholesterol < 40 mg/dL in men and <50 mg/dL in women and/or elevated LDL cholesterol ≥ 135 mg/dL. Waist-to-hip ratio (WHR) was calculated as waist circumference divided by hip circumference. Fat mass index (FMI) was estimated as total body fat divided by height in meters squared (kg/m^2^).

### 4.3. Statistical Analysis

We have performed the statistical procedures using R 4.1.3 software (R Foundation for Statistical Computing, Vienna, Austria). We calculated simple descriptive statistics (means and standard deviations) for continuous variables, while we reported the number of patients and percentages for discrete variables. We used the Shapiro-Wilk test to check the normality of the distribution and then applied the T-test for variables with a normal distribution, while otherwise we used the Wilcoxon rank sum test. We used Fisher’s exact test to analyze differences between proportions and the Spearman’s rank correlation coefficient to describe the associations between serum EGF levels and clinical symptoms. The significance level was set at *p* < 0.05 (two-sided).

## 5. Conclusions

Our data indicate that improvement in negative symptoms due to sarcosine augmentation is not directly mediated by EGF, while effective treatment may promote the production or reduce processes leading to decrease in EGF levels and thus indirectly promote protective processes, including those in which EGF is involved. 

Although our findings indicate a neuroprotective effect of the treatment and confirm the link between neuroprotection and EGF levels, we did not find a direct link between sarcosine action and EGF concentrations, indicating their independent participation in mechanisms of recovery and neuroprotection.

## Figures and Tables

**Figure 1 pharmaceuticals-16-01557-f001:**
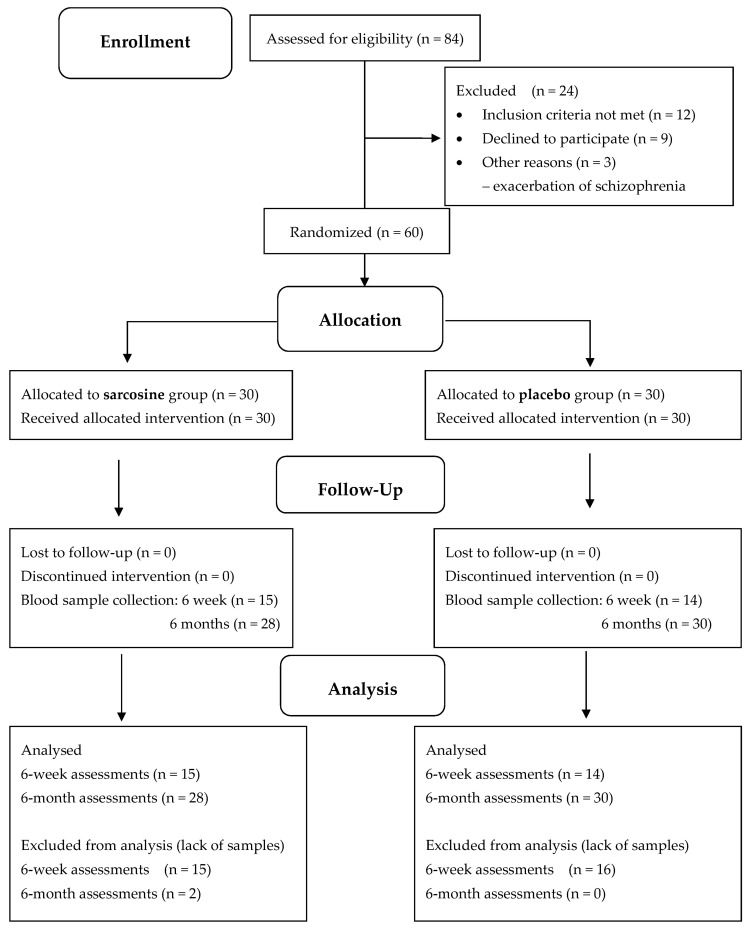
CONSORT diagram for PULSAR study and serum EGF evaluation.

**Figure 2 pharmaceuticals-16-01557-f002:**
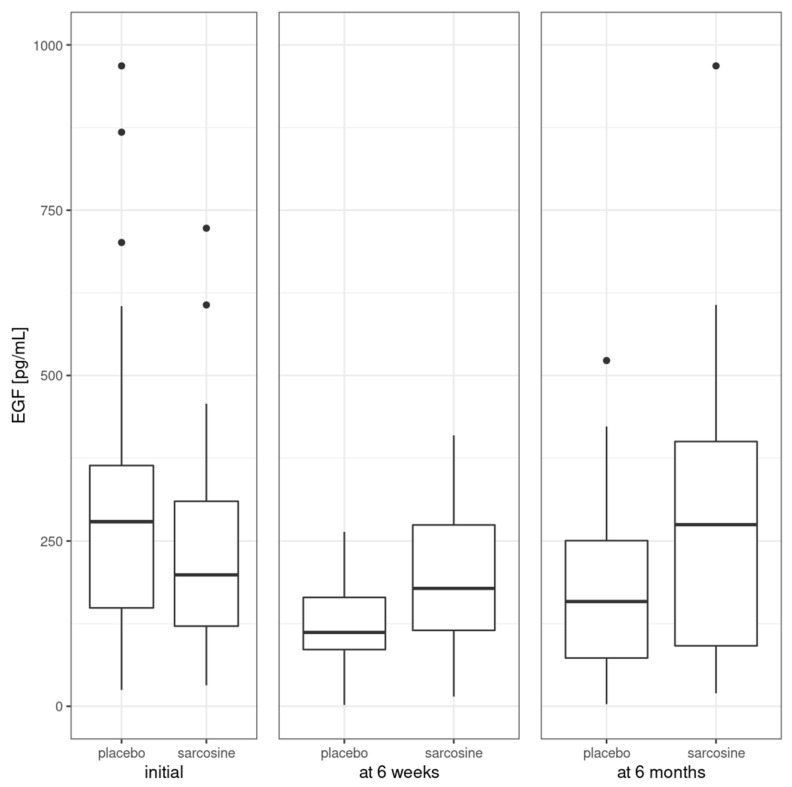
Mean EGF levels in the study groups. Vertical bars represent medians, boxes represent 25th to 75th quartile range, horizontal lines indicate minimum and maximum values and dots represent outliers.

**Figure 3 pharmaceuticals-16-01557-f003:**
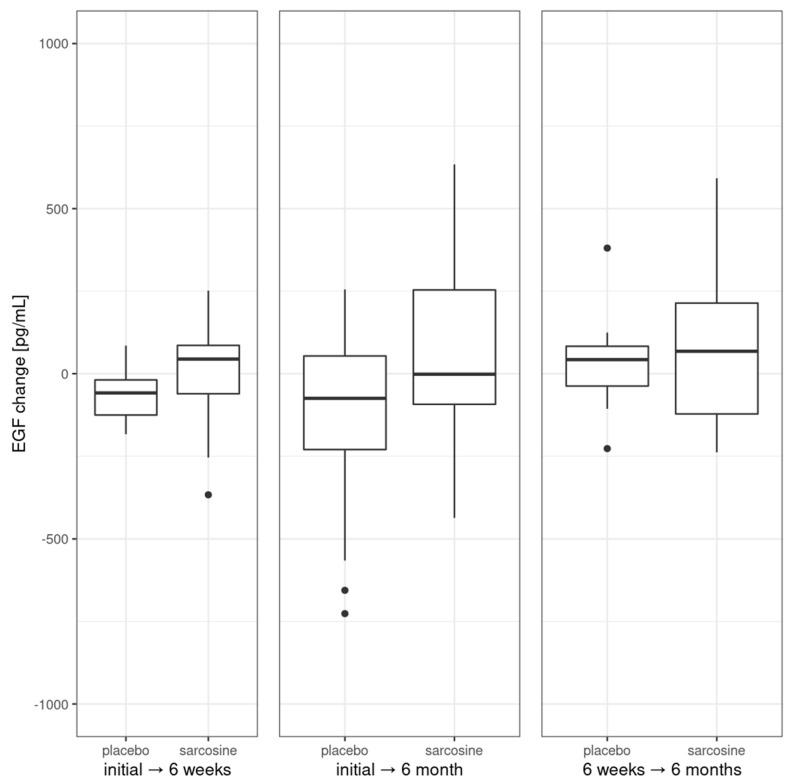
Changes in EGF levels in the study groups. Vertical bars represent medians, boxes represent 25th to 75th quartile range, horizontal lines indicate minimum and maximum values and dots represent outliers.

**Table 1 pharmaceuticals-16-01557-t001:** Characteristics of participants at baseline.

	Sarcosine (*n* = 28)	Placebo (*n* = 30)	*p*
Men	20 (71.4%)	15 (50.0%)	NS
Age (years)	36.7 ± 11.2	40.2 ± 10.1	NS
Smoking	10 (35.7%)	19 (63.3%)	NS
Cardio-metabolic characteristics			
SBP (mm Hg)	125.2 ± 15.9	126.7 ± 16.4	NS
DBP (mm Hg)	75.1 ± 9.4	79.3 ± 9.2	NS
TC (mg/dL)	200.7 ± 30.8	221.2 ± 54.1	NS
HDL (mg/dL)	45.72 ± 18.8	45.5 ± 14.8	NS
LDL (mg/dL)	123.5 ± 30.6	143.6 ± 43.9	NS
TGA (mg/dL)	156.9 ± 86.6	161.3 ± 106.8	NS
FPG (mg/dL)	95.7 ± 13.9	97.6 ± 22.9	NS
TSH (μIU/mL)	1.8 ± 0.9	1.5 ± 0.7	NS
PRL (ng/mL)	31.6 ± 29.1	31.5 ± 36.4	NS
Antihypertensive treatment	4 (14.8%)	7 (23.3%)	NS
Lipid-lowering treatment	1 (3.7%)	2 (6.7%)	NS
Antidiabetic treatment	1 (3.7%)	0	NS
Metabolic syndrome	14 (51.8%)	18 (60.0%)	NS
Dyslipidemia	22 (81.5%)	25 (83.3%)	NS
Impaired fasting glucose	7 (25.9%)	9 (30.0%)	NS
Clinical characteristics			
Treatment duration (years)	14.3 ± 9.1	11.6 ± 5.0	NS
Number of hospitalizations	4.8 ± 5.7	4.2 ± 4.8	NS
Time from last hospitalization (years)	3.1 ± 4.2	4.7 ± 4.6	NS
Number of APs			NS
1	15 (55.6%)	13 (44.9%)	
2	11 (40.7%)	14 (48.3%)	
3	1 (3.7%)	2 (6.9%)	
SGAs	25 (92.6%)	28 (96.5%)	NS
FGAs	3 (11.1%)	8 (26.7%)	NS
Antidepressants	9 (33.3%)	6 (20.7%)	NS
Initial PANSS score	68.2 ± 13.2	72.5 ± 12.5	NS
Positive subscale	9.7 ± 2.6	10.4 ± 3.15	NS
Negative subscale	25.2 ± 5.0	26.1 ± 5.0	NS
General subscale	33.4 ± 7.9	35.9 ± 7.5	NS
Initial CDSS score	3.4 ± 2.9	3.6 ± 2.8	NS
Patients with depression	4 (14.8%)	5 (16.7%)	NS
Body composition			
Weight (kg)	93.6 ± 23.1	86.43 ± 16.4	NS
BMI (kg/m^2^)	34.8 ± 22.6	29.5 ± 4.9	NS
FMI (kg/m^2^)	12.0 ± 8.6	10.5 ± 4.7	NS
Abdominal circumference (cm)	105.7 ± 17.9	103.3 ± 11.9	NS
Waist circumference (cm)	97.3 ± 16.9	96.1 ± 10.7	NS
Hip circumference (cm)	107.7 ± 16.4	105.6 ± 12.4	NS
WHR	0.9 ± 0.1	0.9 ± 0.1	NS
Total body fat (kg)	32.5 ± 17.9	30.4 ± 12.7	NS
Total body fat (%)	32.6 ± 11.5	34.3 ± 11.4	NS
Lean body mass (kg)	61.1 ± 8.9	55.9 ± 11.1	NS
Lean body mass (%)	67.4 ± 11.5	65.7 ± 11.4	NS

Data given as: *n* (%) or mean ± standard deviation. SBP = systolic blood pressure; DBP = diastolic blood pressure; TC = total cholesterol; HDL = high-density lipoproteins; LDL = low-density lipoproteins; TGA = triglycerides; FPG = fasting plasma glucose; TSH = thyroid-stimulating hormone; PRL = prolactin; APs = antipsychotics; SGAs = second-generation antipsychotics; FGAs = first-generation antipsychotics; PANSS = Positive and Negative Syndrome Scale; CDSS = Calgary Depression Scale for Schizophrenia; BMI = body mass index; FMI = fat mass index; WHR = waist-to-hip ratio.

## Data Availability

Data is contained within the article.

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
