# Peer review of "Sarcosine May Induce EGF Production or Inhibit the Decline in EGF Concentrations in Patients with Chronic Schizophrenia (Results of the PULSAR Study)"

_pharmaceuticals, 2023, doi:10.3390/ph16111557_

Round 1

Reviewer 1 Report

Comments and Suggestions for Authors

This paper evaluates the associations between glutaminergic modulation, clinical changes, and peripheral EGF concentrations in patients with chronic schizophrenia. The aim is to determine whether ECG levels in peripheral blood are associated with the severity of schizophrenia symptoms and whether modulation of glutamatergic transmission have impact on symptomatology and ECG concentrations.  I consider that the research topic is original and relevant in this area. It complements the specific gap in this area. The results presented may improve the treatment of patients with chronic schizophrenia and represent a contribution to everyday clinical practice. This is the first study to investigate a similar combination of parameters in subjects with schizophrenia. References are relevant to the study and in the correct style. It is not clear how chronic schizophrenia is defined? In the Discussion chapter, sections 1), 2), and again 1), are not clearly arranged. The conclusion is missing, which is the most important part of the article.

Author Response

Dear Reviewer,

We want to thank you for your comments. According to your questions and notes, we added a description of chronic schizophrenia (in introduction), rearranged the discussion chapter, and summarized the article with conclusions.

Thank you again,

Authors

Reviewer 2 Report

Comments and Suggestions for Authors

The manuscript entitled “Sarcosine may induce EGF production or inhibit the decline in EGF concentrations in patients with chronic schizophrenia (results of the PULSAR study)” (pharmaceutics-2623230) by Pawlak et al provides the results of the effects of Sarcosine on negative symptoms of Schizophrenia and providing the inputs on effect of sarcosine on the serum EGF levels.

The following concerns need to be addressed before accepting the manuscript for publication.

1.      The introduction section needs minor rearrangement, the literature report “Previous studies have considered the severity…” was written after a discussion on the aim of the study. These paragraphs can be interchanged to give readers the best idea of the study.

2.      In the introduction, “schizophrenia is sparse, but authors indicate that EGF” the term authors should be changed to researchers to provide better scientific understanding.

3.      In the same paragraph of the introduction, “The results of several published studies do not provide clear conclusions” needs to be supported with references.

4.      There is an inference for sarcosine has beneficial effects on negative symptoms of schizophrenia, as also mentioned by the authors. But what is the rationale for authors to measure sarcosine-based EGF serum levels?

5.      In the abstract “Our data indicate that improvement in negative symptoms due to sarcosine augmentation is not mediated by EGF, but effective treatment may induce the production/block the decrease in EGF concentration” Why did authors comment on EGF levels and its role in neuroprotection when they mentioned that the study has no role on EGF levels?

6.      Is the figure 1 provided in the manuscript complete? I can see an incomplete version, verify!

7.      The latest references can be used to indicate EGF neuroprotective effects (Ref 31 is from 1983!)

8.      Minor typographical errors like spacings, spelling mistakes, and excess round brackets should be checked for in the entire manuscript.

9.      Language refinement is suggested.

10.  The uniformity in the references should be maintained. All the references should be carefully formatted. For example,

a.       DOI missing for reference 1–4.

b.      In references 4, 7, 24 etc: Page numbers missing.

Comments on the Quality of English Language

Language refinement is suggested

Author Response

Answer to Reviewer 2.

Dear Reviewer,

We want to thank you for your comments. According to your questions and notes, we upgraded the manuscript and prepared answers to your remarks.

  1. The introduction section needs minor rearrangement, the literature report “Previous studies have considered the severity…” was written after a discussion on the aim of the study.

These paragraphs can be interchanged to give readers the best idea of the study.

We have changed it as suggested, thank you.

  1. In the introduction, “schizophrenia is sparse, but authors indicate that EGF” the term authors should be changed to researchers to provide better scientific understanding.

We have changed it as suggested, thank you.

  1. In the same paragraph of the introduction, “The results of several published studies do not provide clear conclusions” needs to be supported with references.

We have included the references in the following sentences, which cover the issues in more detail.

  1. There is an inference for sarcosine has beneficial effects on negative symptoms of schizophrenia, as also mentioned by the authors. But what is the rationale for authors to measure sarcosine-based EGF serum levels?

We added information about the involvement of neuregulin-1, a protein from the EGF family, in the pathogenesis of schizophrenia and dysfunction of the glutamatergic system in this psychosis, which was one of the basis for the analysis of EGF in our study.

  1. In the abstract “Our data indicate that improvement in negative symptoms due to sarcosine augmentation is not mediated by EGF, but effective treatment may induce the production/block the decrease in EGF concentration” Why did authors comment on EGF levels and its role in neuroprotection when they mentioned that the study has no role on EGF levels?

We added the word "directly" to better reflect our interpretation of the results.

  1. Is the figure 1 provided in the manuscript complete? I can see an incomplete version, verify!

We have corrected the graphic view so that the entire figure is visible.

  1. The latest references can be used to indicate EGF neuroprotective effects (Ref 31 is from 1983!)

These references are basic in EGF research, but we also added references form recent years, thank you.

  1. Minor typographical errors like spacings, spelling mistakes, and excess round brackets should be checked for in the entire manuscript.

We upgraded these issues, thank you.

  1. Language refinement is suggested.

We polished the text, thank you.

  1. The uniformity in the references should be maintained. All the references should be carefully formatted. For example,
  2. DOI missing for reference 1–4.
  3. In references 4, 7, 24 etc: Page numbers missing.

We have formatted the entire reference part, thank you.

Thank you again,

Authors

Reviewer 3 Report

Comments and Suggestions for Authors

This manuscript needs to be improved by including the good quality of figs and also need to be included the result and discussion in detailed manner with recent research. 

Author Response

Dear Reviewer,

we want to thank you for your comments. We tried to improve the figures, especially Figure 1, which disintegrated during the text formatting. We also tried to enrich the introduction and discussion with the latest research results, but there is no literature with which we could discuss in detail on the topics we are interested in. We refer to these works as much as we think is possible.

Thank you again,

Authors

Round 2

Reviewer 3 Report

Comments and Suggestions for Authors

The manuscript can be considered for further process.